Production of exopolysaccharide by strains of Lactobacillus plantarum YO175 and OF101 isolated from traditional fermented cereal beverage

Adesulu-Dahunsi Adekemi Titilayo adesuluchemmy@yahoo.com 1
Jeyaram Kumaraswamy 2
Sanni Abiodun Isiaka 1
Banwo Kolawole 1
1 Department of Microbiology, University of Ibadan , Ibadan , Oyo , Nigeria
2 Microbial Resource Division, Institute of Bioresources and Sustainable Development , Imphal , Manipur , India
Conte-Junior Carlos
Electronic publication date: 2018 Oct 10
Publication date: 2018
Volume: 6
Electronic Location ID: e5326
Received 2018 Feb 27; Accepted 2018 Jul 3
Copyright: ©2018 Adesulu-Dahunsi et al.
Copyright year: 2018
Copyright holder: Adesulu-Dahunsi et al.
License: This is an open access article distributed under the terms of the Creative Commons Attribution License, which permits unrestricted use, distribution, reproduction and adaptation in any medium and for any purpose provided that it is properly attributed. For attribution, the original author(s), title, publication source (PeerJ) and either DOI or URL of the article must be cited.
License URL: https://creativecommons.org/licenses/by/4.0/

Keywords: Ogi, Lactobacillus plantarum, Exopolysaccharide, FT-IR, Antioxidant.

Funding: World Academy of Sciences (TWAS) Department of Biotechnology, India This work was funded by the World Academy of Sciences (TWAS) and the Department of Biotechnology, India. The funders had no role in study design, data collection and analysis, decision to publish, or preparation of the manuscript.

==============================
Lactobacillus plantarum YO175 and OF101 isolates from Nigerian traditional fermented cereal gruel ‘ogi’, were investigated on the basis of their capability to produce exopolysaccharide (EPS) on sucrose modified deMan Rogosa Sharpe medium (mMRS). Functional groups analysis of the EPSs produced (EPS-YO175 and EPS-OF101) by Fourier-transform infrared (FT-IR) spectroscopy revealed the presence of –OH, C=O and C–H groups. The chemical composition of EPS-YO175 and EPS-OF101 showed the presence of 87.1% and 80.62% carbohydrates and 1.21% and 1.47% protein. For maximum EPS yield, three significant factors were optimized using central composite design and response surface methodology, the predicted maximum EPS produced was 1.38 g/L and 2.19 g/L, while the experimental values were 1.36 g/L and 2.18 g/L for EPS-YO175 and EPS-OF101. The EPS samples showed strong antioxidant activities in-vitro. The scale-up of the production process of the EPS will find its potential application in food industries.

Introduction

The ability of various lactic acid bacteria (LAB) to produce extracellular long-chain polysaccharides/exopolysaccharides which consists of branched and repeating units of sugars in varied ratios have been reported (Pan & Mei, 2010; Wang et al., 2010; Li et al., 2014; Imran et al., 2016). The EPS produced have immense commercial importance because of the industrially beneficial physico-chemical properties they exhibit and GRAS (generally recognized as safe) status of the LAB from which they are secreted (Surayot et al., 2014). Exopolysaccharide produced by LAB play essential roles in improving the mouth feel, texture, and rheology of fermented food preparations. They also serve as food additives, prebiotics and demonstrate useful physiological effects such as anticarcinogenecity, antitumor, immunomodulating activities and as blood cholesterol-lowering agents in humans (Kim et al., 2010).

Many indigenous foods produced in Nigeria are usually fermented before consumption (Adesulu & Awojobi, 2014). Lactic acid bacteria are commonly isolated from Nigerian indigenous fermented foods and beverages (Banwo, Sanni & Tan, 2013; Sanni & Adesulu, 2013; (Adesulu-Dahunsi, Sanni & Jeyaram, 2017). Ogi is a popular non-alcoholic fermented cereal beverage that is processed by natural fermentation with the dominance of LAB. It forms a staple food of the people in West Africa, especially among the south-western Nigerian where it serves as weaning food in infants and/or breakfast beverage among the adults. Exopolysaccharides produced by some of the LAB isolated from these fermented food products have been documented to improve the food texture and quality. Exopolysaccharides producing LAB such as Streptococcus thermophilus, Lactobacillus delbrueckii subsp. bulgaricus, Lactococcus lactis subsp. cremoris isolated from dairy products and fermented milk have been extensively studied (Patel & Prajapati, 2013).

Different species of LAB, especially Lactobacillus plantarum have been reported to produce EPS (Wang et al., 2010; Imran et al., 2016). L. plantarum perform important fermentative roles during Nigerian traditional food preparation and provides positive health impacts which are strain specific, they exhibit an outstanding effect on the flavour and texture of these foods, with specific metabolic and technological properties, such as production of EPS (Adesulu-Dahunsi et al., 2017). Recently, researchers have reported that EPS produced from LAB species have antioxidant activities and are non-toxic; these characteristics are of great importance and may replace the synthetic antioxidants (Li et al., 2013; Zhang et al., 2013; Abdhul et al., 2014). Exopolysaccharides produced by Lactobacillus species improves sourdough properties by aiding water absorption, improving its structure, thereby prolonging shelf life of the fermented foods. Few works have been reported on EPS producing ability of LAB strains isolated from cereals-based fermented food (Torres-Rodríguez et al., 2014; Adesulu-Dahunsi et al., 2018; Adesulu-Dahunsi, Jeyaram, & Sanni, 2018).

Optimization of the growth factors and media composition are criteria to be considered for maximal EPSs production by LAB strains (Zajseket, Gorsek &Kolar, 2013; Adesulu-Dahunsi, Sanni & Jeyaram, 2018). The statistical designs used in this study to determine the optimal conditions for the EPS production are central composite design (CCD) and response surface methodology (RSM). The objective of this study was to improve the production of EPS from L. plantarum strains and to evaluate the antioxidant activities in vitro. The factors affecting the production of EPS from L. plantarum YO175 and OF101 were analyzed, and three significant variables (cultivation time, pH and sucrose concentration) were chosen to optimize the production conditions using CCD and RSM. In addition, the in vitro antioxidant assays of the EPS were also evaluated.

Material and Methods

Microorganisms and chemicals

Two EPS-producing LAB strains isolated from ogi (Nigerian indigenous fermented cereal gruel from yellow and white maize varieties) were identified according to their biochemical characteristics and 16S rRNA gene sequencing as L. plantarum YO175 and L. plantarum OF101 (GenBank Accession numbers KU892395 and KU892393). DPPH, trichloroacetic acid (TCA), trifluoroacetic acid (TFA), Folin-Ciocalteu reagent, bovine serum albumin (BSA), phenol, concentrated sulfuric acid, methanol, ferric chloride, potassium ferric cyanide, pyrogallol, ascorbic acid, glucose, galactose, rhamnose, xylose, fructose and ribose (Sigma Chemical Ltd., St. Louis, USA).

Extraction and purification of EPS

The isolation and partial purification of the EPS samples were performed as previously described by Savadogo et al. (2004), and modified by Adesulu-Dahunsi et al. (2018) with mMRS broth as the cultivation medium as it support the growth of L. plantarum species and MRS as growth medium for all LAB species. Briefly, L. plantarum YO175 and L. plantarum OF101 were propagated in 1,000 mL MRS modified sucrose broth (glucose was replaced with sucrose) and incubated for 24 h in an incubator shaker, the L. plantarum cultures were heated at 100 °C for 10 min, then centrifuged at 12,000× g for 15 min to remove the cells, the supernatants was precipitated with double volume ice-cold ethanol, shaken vigorously and centrifuged at 5,000× g for 30 min at 4 °C. The pellets obtained were dried to a constant weight in an oven at 50 °C. The resulting pellets were mixed with ultrapure water, the EPS slurry was precipitated with double volume of cold ethanol, this step was repeated three times in order to eliminate any cells debris that may be present. The EPS was mixed with minimal water and dialyzed against distilled water using 10 kDa membranes for 48 h at 4 °C with the changing of the water twice daily. The partially purified EPS obtained from dialysis was frozen at −20 °C in deep freezer. The frozen EPS was covered with parafilm and were lyophilized for 2 days and the resulting EPS samples were preserved for further characterization.

Molecular mass determination

The molecular mass of the EPS-YO175 and EPS-OF101 were determined on an Agilent 1100HPLC system equipped with a TSK-GEL G3000SWxl column (7.5 × 300 mm, Tosoh Corp., Tokyo, Japan) and a refractive index detector (RID). The column was eluted with 0.1 M Na2SO4 solution at a flow rate of 0.8 mL min−1. Molecular mass was estimated from the standard graph which was plotted using standard dextrans (Sigma–Aldrich, USA) (Pan & Mei, 2010).

Chemical composition analysis

Total carbohydrates, total soluble protein content of the EPS samples, and lactic acid (LA) produced by the L. plantarum strains were determined.

The EPS samples were estimated using phenol-sulphuric acid method (Dubois et al., 1956). 1 mL of the EPS sample was added to 1 ml of 5% cold phenol and 5 mL concentrated sulphuric acid placed in an ice bath, then incubated at room temperature for 20 min and the absorbance of the samples at 490 nm was taken using a spectrophotometer.

Estimation of the protein content was performed according to Lowry et al. (1951). To 1 mL EPS sample (s), 5 mL (2% Na2CO3 in 0.1 N NaOH and 0.5% CuSO4 in 1% potassium sodium tartrate in 50:1) was added and vortexed. The mixture was incubated at ambient temperature for 10 min, 0.5 mL 1N Folin-Ciocalteau reagent was added and mixed and was kept in dark for 20 min after which the colour was measured at 660 nm.

The production of LA in MRS broth by the L. plantarum strains was estimated. To 1 mL of the culture supernatants, 0.05 mL of 4% CuS04 and 6 mL of concentrated sulphuric acid was added and mixed well. The mixture was then incubated in boiling water bath for 10 min, 100 µL of p-hydroxydiphenyl was added and kept in room temperature for 30 min. The absorbance was measured at 560 nm (Salvucci, LeBlanc & Perez, 2016).

Analysis of monosaccharide composition

The monosaccharide composition of the purified EPS samples was determined by thin layer chromatography (TLC) via aluminum plates coated with silica gel and high-performance liquid chromatography (HPLC) after hydrolysis of the EPS. Five milligram 5 mg of the purified EPS sample (s) was dissolved in 0.5 mL MilliQ water and hydrolyzed in 0.5 mL of 6 N trifluoroacetic acid (TFA) at 100 °C for 3 h. The hydrolysate was evaporated to dryness at 50 °C. Five microlitres (5 µL) of the EPS samples were spotted onto a silica gel coated aluminum TLC plates. The mixed solvent (n-butanol, ethanol, and water (50:30:20 v/v/v) were used for separation of carbohydrates and standards, the fractions were visualized on the TLC plates, after dipping it in anisaldehyde-sulphuric acid reagent and heating the plates at 110 °C for 30 min. The EPS was also analyzed with HPLC system (Agilent 1100) equipped with Aqueous GPC start up Kit column and eluted with distilled water at a flow rate of 1.0 mL/min at 20 °C. The separated components were monitored by a refractive index detector.

Analysis of functional groups

The infrared analysis of purified EPS from the two L. plantarum strains was carried out using an FT-IR spectrophotometer (Thermo Nicolet, USA) in the spectrum ranges of 400–4,000 cm−1 for the detection of functional groups present in the samples.

Preliminary screening of cultivation condition and media composition for EPS production

The optimal media composition and cultivation condition for EPS production in MRS broth were screened. The media components, carbon source (20 g/L), organic nitrogen sources (25 g/L), and inorganic nitrogen sources (2 g/L), were substituted independently into the media and by keeping other components constant at different cultivation times (12, 24, 36, 48, 60, 72, 84 and 96 h), initial pH of the media (6, 6.5, 7, 8) and different temperatures (20, 25, 30, 37 and 45 °C).

In vitro determination of antioxidant properties

The in-vitro antioxidant properties of the purified EPS samples (at 0.5–4 mg/mL concentration levels) were performed using standard methods.

The DPPH (1,1-diphenyl-2-picryl-hydrazyl) radical scavenging was measured according to the method of Rai et al. (2011). A 2.0 mL deionized water and 2.0 mL DPPH solution (0.16 mM) were added to 1.0 mL EPS samples, the mixture was incubated at 37 °C in a dark room for 30 min. Absorbance (at 517 nm) was measured against the blank. For the positive control, methanol was replaced with DPPH. The DPPH radical%scavenging activity=1−As−Ab∕Ac×100

As = Absorbance of the EPS sample (s); Ab = Absorbance of blank; Ac = Absorbance of control.

Superoxide scavenging activity of the EPS samples was performed according to Balakrishnan et al. (2011) published protocol. The mixture contained 0.3 mL of the EPS samples, 2.6 mL (50 mM) phosphate buffer at pH 8.2 and 90 µL of 3 mM pyrogallol (dissolved in 10 mM HCl). Then the absorbance (at 325 nm) was taken at 0 min and 10 min. Superoxide scavenging activity%=1−A10−A0∕C10−C0×100

A0 absorbance of EPS sample (s) at 0 min; A10 absorbance of EPS sample (s) at 10 min; C0 absorbance of control at 0 min; C10 absorbance of control at 10 min.

To measure the reducing power potential of the EPS samples, the following mixture; 100 µL of the EPS sample, 900 µL of phosphate buffer (0.2 M, pH 6.6) and 900 µL of 1% potassium ferric cyanide were incubated at 50 °C for 20 min. Nine hundred microliters of TCA (10%) was mixed with the solution and centrifuged at 5,000× g for 15 min, then 900 µL each of the supernatant solution, distilled water and 0.1% ferric chloride were mixed together. The solution was mixed together and the absorbance (700 nm) was taken (Balakrishnan et al., 2011).

The hydroxyl radical scavenging activity of the EPS samples was measured with the Fenton reaction. One milliliter (1 mL) of the EPS samples was added to the reaction mixture containing 1.0 mL of brilliant green (0.435 mM), 2.0 mL of FeSO4 (0.5 mM) and 1.5 mL of H2O2 (3.0%, w/v) and was incubated at 37 °C for 20 min, and the absorbance (at 624 nm) was then measured (Balakrishnan et al., 2011). The hydroxyl radical scavenging activity%=As−A0∕A−A0×100

AS = Absorbance of the sample; A0 = Absorbance of the control; A = Absorbance of deionized water without the sample and Fenton reaction.

Statistical analysis

All experiments were performed in triplicate and the results represented by their mean ± SD. Tests of significant differences were determined by Duncan’s Multiple Range Test at (P < 0.05). For maximum EPS production, Design-Expert software version 8.0.7.1 (Stat-Ease Inc., Minneapolis, USA) was used for the experimental designs and regression analysis of the experimental data. Statistical analysis of the model was performed to evaluate the analysis of variance (ANOVA) and and its statistical significance was determined by F-test.

Results

Molecular mass of the EPS

Based on the calibration curve of the elution retention time of various dextran standards used, the molecular mass of EPS-YO175 and EPS-OF101 was estimated to be 1.2 × 106 Da and 4.4 × 105 Da.

Chemical composition analysis of the EPS

Total carbohydrate content in EPS-YO175 was 87.1% and 80.62% for EPS-OF101. The total soluble protein showed low protein content of approximately 1.21% and 1.47% for EPS-YO175 and EPS-OF101. The lactic acid produced by L. plantarum YO175 and OF101 was 13.6 ± 0.1 mg/mL and 11.4 ± 0.15 mg/mL, respectively.

Analysis of monosaccharide composition

The TLC analysis of the EPS samples was compared with the sugar standards, and their retention time values revealed that EPS-YO175 composed of glucose and galactose and EPS-OF101 showed only glucose (Fig. S1).

The retention time of the EPS samples was compared with the reference standards in the HPLC analysis; this also confirmed that the EPS samples contained glucose and galactose (Fig. 1).

Figure 1 HPLC chromatograms of the EPS samples and standards.

(A) standards, (B) EPS-YO175, (C) EPS-OF101. The peaks correspond to glucose (peak 1), xylose (peak 2), galactose (peak 3), fructose (peak 4), rhamnose (peak 5).

Analysis of functional groups of the EPS

The FTIR spectrum of the EPS samples revealed that the polysaccharides contained a significant number of O-H group representing vibration of the hydroxyl groups of carbohydrate showed by broad stretching in the region 3,288 cm−1 and 3,276 cm−1. The stretching bands around the 2,924 cm−1 and 2,898 cm−1 region were due to C–H stretching vibration. The absorptions around 1,655 cm−1 and 1,649 cm−1 were due to stretching vibration of carbonyl group (C=O). The bands observed around 1,159 cm−1 in EPS-YO175 and 1,150 cm−1 in EPS-OF101 were attributed to the vibration of C-O-C bond (Fig. 2).

Figure 2 FTIR spectrum of EPS produced.

(A) Lactobacillus plantarum YO175 (B) Lactobacillus plantarum OF101.

Preliminary screening of cultivation condition and media composition

Among the different carbon sources, the highest EPS yield was obtained for the sucrose supplemented MRS broth in L. plantarum YO175 (1.59 ± 0.06 g/L) and OF101 (1.07 ± 0.01 g/L). Also at 20 g/L sucrose concentration, the EPS yield obtained for both L. plantarum strains were; 1.64 ± 0.11 and 1.05 ± 0.02 g/L respectively. Yeast extract was the most effective among the nitrogen sources screened, as the optimal EPS yield obtained were 1.63 ± 0.05 g/L and 1.01 ± 0.02 g/L in L. plantarum YO175 and OF101. Ammonium sulphate showed optimal EPS yield in both strain; L. plantarum YO175 (1.56 ± 0.02 g/L) and OF101 (1.00 ± 0.02 g/L). The optimal temperature and pH for the EPS production was 30 °C and pH 7.0 respectively with the corresponding EPS (g/L) yield (1.56, 1.03; and 1.72; 1.10) (Table 1). The dry EPS samples are shown in Fig. S2.

Table 1 Effect of carbon sources, nitrogen sources, and cultivation conditions on exopolysaccharide production.

The different carbon sources, organic and inorganic nitrogen and media composition, cultivation conditions were investigated.

Medium sources	EPS yield (g/L)	
	EPS-YO175	EPS-OF101	
Carbon sources (20 g/L)			
Glucose	0.94 ± 0.10	0.62 ± 0.44	
Sucrose	1.59 ± 0.10	1.07 ± 0.26	
Lactose	1.48 ± 0.14	1.03 ± 0.17	
Galactose	0.87 ± 0.26	0.41 ± 0.26	
Sucrose concentrations (g/L)			
10	1.13 ± 0.20	0.90 ± 0.62	
20	1.64 ± 0.44	1.05 ± 0.44	
30	1.52 ± 0.26	0.71 ± 0.56	
40	1.45 ± 0.75	0.62 ± 0.17	
50	1.10 ± 0.00	0.36 ± 0.10	
Organic nitrogen (25 g/L)			
Yeast extract	1.63 ± 0.07	1.01 ± 0.11	
Beef extract	1.15 ± 0.22	0.82 ± 0.10	
Tryptone	1.48 ± 0.31	0.83 ± 0.24	
Peptone	1.61 ± 0.01	0.77 ± 0.20	
Inorganic nitrogen (2 g/L)			
Ammonium sulphate	1.56 ± 0.10	1.00 ± 0.00	
Ammonium nitrate	1.51 ± 0.00	0.80 ± 0.00	
Ammonium chloride	1.48 ± 0.21	0.71 ± 0.10	
Tri-ammonium citrate	1.37 ± 0.32	0.75 ± 0.12	
Sodium nitrite	1.30 ± 0.10	0.68 ± 0.26	
Potassium nitrate	1.15 ± 0.24	0.72 ± 0.18	
Temperature (°C)			
20	0.53 ± 0.20	0.20 ± 0.01	
25	0.77 ± 0.01	0.31 ± 0.08	
30	1.56 ± 0.10	1.03 ± 0.11	
37	1.50 ± 0.18	0.80 ± 0.00	
45	0.46 ± 0.24	0.10 ± 0.10	
Cultivation time (h)			
12	0.30 ± 0.06	0.10 ± 0.04	
24	1.70 ± 0.08	0.76 ± 0.10	
36	1.66 ± 0.24	1.00 ± 0.12	
48	1.68 ± 0.20	1.08 ± 0 22	
60	1.50 ± 0.10	1.02 ± 0.24	
72	1.38 ± 0.07	0.72 ± 0.11	
84	1.17 ± 0.04	0.66 ± 0.22	
96	1.00 ± 0.30	0.60 ± 0.32	
pH			
6.0	1.51 ± 0.11	0.65 ± 0.01	
6.5	1.60 ± 0.00	1.01 ± 0.01	
7.0	1.72 ± 0.24	1.10 ± 0.02	
8.0	1.50 ± 0.33	0.82 ± 0.07	
Notes.

Values are means of three independent experiments (mean ± SD).

Response surface optimization for enhanced EPS production

A design model with 20 runs in 1 block was performed and the variables were tested at four levels (Tables 2A and 2B). The statistical significance of the experimental data was checked using Fisher’s statistical test for ANOVA, the 3D graphs were also designed. Tables 3A and 3B showed that the model p-values of 0.0053 and 0.0010 for EPS-YO175 and EPS-OF101 were significant. Also, the Model F-values of 5.88 and 8.88 for EPS-YO175 and EPS-OF101 imply that the models were significant. In EPS-YO15 and EPS-OF101, the lack-of-fit term was greater than 0.05 which is non-significant relative to the pure error.

A Table 2A The central composite experimental design matrix showing the predicted and experimental values of exopolysaccharide from Lactobacillus plantarum YO175.

The CCD of the EPS samples.

Run	A	B	C	EPS Yield (g/L)	
	Cultivation time (hr)	Sucrose concentration (g)	pH	Experimental (E1)	Predicted	
1	45.48	24.00	6.99	1.28	1.22	
2	46.80	23.09	6.87	1.29	1.25	
3	47.95	21.09	6.51	1.30	1.31	
4	46.18	20.34	7.50	1.34	1.32	
5	45.74	20.00	6.42	1.35	1.33	
6	45.96	19.92	7.44	1.32	1.34	
7	45.93	20.34	8.00	1.31	1.30	
8	46.65	20.64	7.71	1.33	1.31	
9	46.05	19.62	6.94	1.34	1.28	
10	48.50	23.00	7.40	1.36	1.38	
11	45.89	19.88	6.74	1.32	1.31	
12	45.86	16.50	7.97	1.28	1.28	
13	45.67	20.11	7.80	1.26	1.31	
14	46.18	20.13	7.88	1.30	1.32	
15	46.82	20.29	8.01	1.33	1.34	
16	46.44	20.85	7.73	1.35	1.33	
17	45.94	20.04	7.00	1.30	1.27	
18	46.03	19.87	7.98	1.29	1.30	
19	45.81	19.71	7.90	1.27	1.31	
20	49.14	20.26	6.86	1.31	1.32	

B Table 2B The central composite experimental design matrix showing the predicted and actual values of exopolysaccharide from Lactobacillus plantarum OF101.

The CCD of the EPS samples.

Run	A	B	C	EPS Yield (g/L)	
	Cultivation time (hr)	Sucrose concentration (g)	pH	Experimental (E1)	Predicted	
1	47.00	20.00	7.00	2.17	2.16	
2	49.00	16.00	8.00	2.12	2.10	
3	49.00	24.00	8.00	2.09	2.10	
4	49.00	24.00	6.00	2.06	2.10	
5	45.00	24.00	6.00	2.04	2.05	
6	49.00	16.00	8.00	2.10	2.12	
7	45.00	16.00	7.76	2.08	2.14	
8	45.00	16.00	6.70	2.06	2.10	
9	45.29	24.00	7.47	2.13	2.14	
10	45.84	18.92	6.98	2.14	2.15	
11	47.64	23.76	7.72	2.14	2.10	
12	45.78	21.28	6.60	2.15	2.06	
13	48.16	18.03	7.24	2.11	2.10	
14	47.70	23.50	7.80	2.18	2.19	
15	47.65	17.10	7.81	2.13	2.15	
16	47.10	23.34	7.40	2.12	2.10	
17	47.16	22.89	7.95	2.15	2.15	
18	47.67	19.24	7.94	2.12	2.10	
19	45.11	17.10	6.70	2.09	2.08	
20	47.88	23.34	6.42	2.04	2.00	

A Table 3A Analysis of Variance (ANOVA) of quadratic model for production of exopolysaccharide in Lactobacillus plantarum YO175.

The ANOVA table generated from the RSM analysis.

Source	Sum of squares	df	Mean square	F value	p-value prob > F		
Model	9.905	9	7.21	5.88	0.0053	Significant	
A-Cultivation time	4.740	1	4.740	2.53	0.1425		
B-Sucrose conc.	1.511	1	1.511	8.08	0.0175		
C-pH	7.389	1	7.389	3.95	0.0749		
AB	1.125	1	1.125	0.60	0.4559		
AC	1.013	1	1.013	5.41	0.0423		
BC	3.125	1	3.125	1.67	0.2252		
A2	2.500	1	2.500	13.37	0.0044		
B2	1.870	1	1.870	10.02	0.0101		
C2	3.369	1	3.369	13.37	0.0044		
Residual	1.870	10	1.870				
Lack of fit	3.369	5	3.369	0.22	0.9391	Not significant	
Pure error	1.533	5	1.04				
Cor total	0.012	19					
Notes.

R-squared = 0.8412; Adequate precision = 8.558.

B Table 3B Analysis of Variance (ANOVA) of quadratic model for production of exopolysaccharide in Lactobacillus plantarum OF101.

The ANOVA table generated from the RSM analysis.

Source	Sum of squares	df	Mean square	F value	p-value prob > F		
Model	0.10	9	0.04	8.88	0.0010	Significant	
A-Cultivation time	4.663	1	4.663	3.65	0.0852		
B-Sucrose conc.	0.010	1	0.010	7.88	0.0186		
C-pH	4.967	1	4.967	3.89	0.0770		
AB	2.813	1	2.813	2.20	0.1688		
AC	3.125	1	3.125	0.24	0.6317		
BC	1.012	1	1.012	0.79	0.3944		
A2	3.982	1	3.982	3.12	0.1080		
B2	0.078	1	0.078	61.20	<0.0001		
C2	0.13	1	0.13	0.27	0.6167		
Residual	0.013	10	0.013				
Lack of fit	0.013	6	0.013	70.35	0.6252	Not-significant	
Pure error	1.200	4	0.80				
Cor total	0.11	19					
Notes.

R-squared = 0.8888; Adequate precision = 10.789.

The developed regression model equations describing the relationship between the EPS yield (Y) and the coded values of independent factors; cultivation time (hr) (A), sucrose concentration (g) (B), pH (C) and their corresponding interactions is described below: (1) YEPS−Y O175=2.15+5.891E−003A+0.011B−7.356E−003C−3.750E−003AB+0.011AC+6.250E−003BC−0.013A2−0.011B2−0.013C2

(2) YEPS−OF101=1.35−0.018A+0.027B+0.019C+0.019AB+6.250E−003AC−0.011B−0.017A2−0.074B2−4.874E−003C2

where Y implies EPS yield (g/L).

Three-dimensional response surface plots and contour of EPS yield from L. plantarum YO175 (Figs. 3A–3C) and OF101 (Figs. 4A–4C) illustrate the interactions between the three variables. The curvatures’ nature of 3D surfaces gave good interaction between sucrose concentration and cultivation time, pH and cultivation time, pH and sucrose concentration.

Figure 3 Response surface three-dimensional plots and corresponding contour plots of the three significant variables on EPS yield for Lactobacillus plantarum YO175.

(A) Sucrose concentration and cultivation time (B) pH and cultivation time (C) pH and sucrose concentration.

Figure 4 Response surface three-dimensional plots and corresponding contour plots of the three significant variables on EPS yield for Lactobacillus plantarum OF101

(A) Sucrose concentration and cultivation time (B) pH and cultivation time (C) pH and sucrose concentration.

The optimal values of the independent factors selected for the production of EPS were obtained by solving the regression equation (Eqs. (1)–(2)) using the Expert Design 8.0.7.1 software package. The optimal values of the tested variables of L. plantarum YO175 were; cultivation time, 48.50 h; sucrose concentration, 23.00 g/L, and pH, 7.40. Under these conditions, the maximum predicted yield of EPS was 1.38g/L and its experimental yield was 1.36 g/L. For L. plantarum OF101, the optimal values of the test variables were cultivation time, 48.00 h; sucrose concentration, 23.50 g/L and pH, 7.50 and the maximum predicted yield was 2.19 g/L and its actual experimental value was 2.18 g/L.

In vitro determination of antioxidant properties

DPPH free radical scavenging activity

The DPPH scavenging activity observed in the EPS samples and ascorbic acid at concentration of 0.5 mg/mL was lower than the ones at 4 mg/mL. At 4 mg/mL, the scavenging activity for the ascorbic acid was significantly higher than those found in L. plantarum YO175 (56.9%) and OF101 (51.2%) respectively (Table 4).

Table 4 In vitro antioxidant activities of the exopolysaccharides from L. plantarum YO175 and OF101.

The antioxidant activities of the EPS samples.

Concentrations (mg/mL)	Ascorbic acid	EPS-YO175	EPS-OF101	
	DPPH scavenging activity (%)	
0.5	38.7 ± 0.21c	22.3 ± 0.79b	18.9 ± 0.66a	
1.0	47.0 ± 0.20c	28.3 ± 0.82b	21.3 ± 0.76a	
1.5	56.1 ± 0.00c	35.3 ± 0.30b	27.4 ± 1.83a	
2.0	60.9 ± 0.40c	39.1 ± 0.23b	32.8 ± 0.17a	
2.5	64.7 ± 1.47c	46.5 ± 0.95b	40.8 ± 1.56a	
3.0	72.1 ± 0.90c	50.7 ± 0.44b	47.6 ± 0.56a	
3.5	78.3 ± 1.74c	54.0 ± 0.00b	49.8 ± 0.20a	
4.0	82.1 ± 1.23c	56.9 ± 1.77b	51.3 ± 0.35a	
	Superoxide scavenging activity (%)	
0.5	37.9 ± 0.36b	43.2 ± 0.53c	23.0 ± 0.00a	
1.0	51.3 ± 0.70b	55.7 ± 0.36c	27.8 ± 0.35a	
1.5	57.4 ± 0.80b	62.1 ± 0.56c	31.4 ± 0.35a	
2.0	64.3 ± 0.53b	69.8 ± 0.79c	35.7 ± 0.70a	
2.5	69.1 ± 1.05b	77.6 ± 0.70c	38.9 ± 1.08a	
3.0	74.2 ± 0.35b	82.6 ± 0.56c	41.8 ± 0.20a	
3.5	80.4 ± 0.53b	86.3 ± 0.70c	42.7 ± 0.45a	
4.0	83.1 ± 0.17b	89.4 ± 0.35c	45.2 ± 0.20a	
	Reducing power activity (Abs 700 nm)	
0.5	0.33 ± 0.62b	0.17 ± 0.02a	0.11 ± 0.00a	
1.0	0.55 ± 0.00b	0.21 ± 0.02a	0.19 ± 0.01a	
1.5	0.67 ± 0.04b	0.25 ± 0.03a	0.23 ± 0.01a	
2.0	0.71 ± 0.04b	0.28 ± 0.03a	0.24 ± 0.03a	
2.5	0.75 ± 0.02b	0.31 ± 0.02a	0.26 ± 0.04a	
3.0	0.82 ± 0.03c	0.37 ± 0.01b	0.28 ± 0.02a	
3.5	0.87 ± 0.11b	0.39 ± 0.04a	0.31 ± 0.02a	
4.0	0.91 ± 0.03c	0.41 ± 0.03b	0.34 ± 0.01a	
	Hydroxyl radical scavenging activity (%)	
0.5	38.7 ± 0.69c	26.8 ± 0.70b	22.0 ± 0.00a	
1.0	49.7 ± 0.61c	37.6 ± 0.36b	27.0 ± 0.35a	
1.5	57.4 ± 0.87c	41.3 ± 0.61b	31.2 ± 1.27a	
2.0	64.3 ± 0.61c	45.8 ± 0.53b	37.9 ± 0.10a	
2.5	69.1 ± 0.27c	53.2 ± 1.59b	44.7 ± 0.62a	
3.0	74.2 ± 0.82c	57.1 ± 0.17b	49.2 ± 0.82a	
3.5	80.4 ± 0.60c	62.3 ± 0.70b	50.1 ± 0.17a	
4.0	83.1 ± 1.39c	66.0 ± 0.00b	52.3 ± 0.30a	
a,b,cMeans in the same column with different superscript letters represent significant difference (P < 0.05) by Duncan’s post hoc comparisons.

Values are means of three independent experiments (mean ± SD).

Superoxide scavenging activity

The superoxide scavenging activity of purified EPS samples and ascorbic acid are shown in Table 4. As the concentration increases from 0.5 mg/mL to 4 mg/mL the scavenging effects also increases; from 23%–45.3% in EPS-OF101, 37.9%–83.1% in ascorbic acid and 43.2%–89.4% in EPS-YO175.

Reducing power activity of EPS

The reducing power of the EPS sample (s) and ascorbic acid increased with the increase in the concentration levels from 0.5 mg/mL to 4 mg/mL. The reducing power of the ascorbic acid, EPS-YO175 and EPS-OF101 at 4 mg/mL concentration are; (0.91 > 0.41 > 0.34), no significant differences was observed between the EPS samples, but the result obtained for the ascorbic acid showed that there was significant different from EPS-YO175 and OF101 (Table 4).

Hydroxyl radical scavenging activity

The hydroxyl radical generated by the Fenton reaction in the system was scavenged by the EPS-YO175, EPS-OF101 and ascorbic acid. Their scavenging effects are shown in Table 4. The two EPS samples exhibited moderate scavenging effect against hydroxyl radical, and EPS-YO175 showing stronger scavenging effect than EPS-OF101. However, ascorbic acid showed higher hydroxyl radical scavenging activity and was significantly different from the EPS samples. At a concentration of 4 mg/mL, the scavenging activity for EPS-YO175, EPS-101, and ascorbic acid was 66.0%, 52.3% and 83.1% respectively.

Discussion

Exopolysaccharides produced by strains of Lactobacillus plantarum YO175 and OF101 isolated from traditional fermented cereal beverage on MRS sucrose modified media were investigated. The chemical composition analysis showed that EPS samples contained significant amounts of carbohydrate content and relatively low protein. High carbohydrate content from LAB-EPS has been documented by several researchers (Li et al., 2014; Wang et al., 2015; Imran et al., 2016). Liu et al. (2002) also reported protein content as low as 2.3% from EPS obtained from the fermentation of kefir grains. Lactic acid is the major metabolic end product of carbohydrate fermentation in LAB. These LAB strains were found to produce LA, and therefore they possess the ability to lower pH in food which may result in the development of desirable organoleptic properties and inhibition of the pathogenic microorganisms in food, thereby ensuring safety and stability of the final product.

The analysis of the EPS monosaccharide composition revealed glucose and galactose monomers which indicates that EPS-YO175 is an heteropolysaccharide and EPS-OF101 is an homopolysaccharide. Our result is in agreement with Imran et al. (2016) who reported that the monomer composition in EPS samples can vary among different strains of the same species. Tallon, Bressollier & Urdaci (2003) reported that L. plantarum EP56 is composed of glucose and galactose. Other researchers have also reported that EPS from L. plantarum are composed of different sugar moieties (Ismail & Nampoothiri, 2010; Imran et al., 2016).

FT-IR is a useful tool for determining the functional groups in EPS (Wang et al., 2008). The functional groups present in the two EPS samples as determined by FT-IR spectral analysis showed absorption bands of polymeric structure(s). The functional groups with the vibration frequencies when compared with the FT-IR spectra analysis of the other polysaccharides reported in the literature confirmed that the two EPS samples are carbohydrates (Wang et al., 2010). Similar FTIR peak range was observed for L.plantarum YW11 isolated from Tibet kefir and L. helveticus MB2-1 isolated from say ram ropy fermented milk (Li et al., 2014; Wang et al., 2015).

The amounts of EPS produced by microorganisms depend solely on the cultivation conditions and media composition (Wang et al., 2010). Carbon and nitrogen sources, cultivation time, temperature and pH have been reported to influence growth and production of EPS by LAB (Gandhi, Rayand & Patel, 1997). Ismail & Nampoothiri (2010) and Kanmani et al. (2011) reported that at 35 °C and pH ranging between 6.5 and 7.0, maximum EPS production was obtained for the EPS producing Lactobacillus and Streptococcus species. The optimum range of cultivation conditions obtained during the production of EPS was similar to those reported by other researchers (Hallemeersch, De Baet & Vandamme, 2002; Sarwat, Aman & Ahmed, 2008; Wang et al., 2015; Adesulu-Dahunsi, Sanni & Jeyaram, 2018). The presence of growth factors such as large quantities of amino acids and short peptides in the yeast extract resulted to enhance EPS production. Liu et al. (2009) also observed that in the presence of yeast extract, high EPS was produced by Paenibacillus polymyca EJS-3. The response surface optimization for enhanced EPS production using RSM and CCD gave good agreement between the experimental and predicted values. This implied that the mathematical models were suitable for the simulation of EPS production in the present study. Many researchers have reported varied amounts of EPS produced by Lactobacillus plantarum species (Zhang et al., 2013; Li et al., 2014).

The in vitro determination of the antioxidant activities increased with increase in their concentrations, no significant difference was observed between the EPS samples. Also, the activities increased in a dose dependent manner as the EPS concentrations increased. Ascorbic acid was found to have higher antioxidant activities when compared with the two EPS samples. The antioxidant potentials displayed by the EPS samples in this study was similar to those reported by other researchers (Sun et al., 2012; Ye et al., 2012; Zhang et al., 2013).

Conclusion

In the present work, the optimization of growth parameters using the statistical methodologies of CCD and RSM for enhanced EPS production by L. plantarum strains (YO175 and OF101) isolated from ogi, a Nigerian indigenous fermented food, was studied. The in vitro antioxidant assays showed that the EPS produced by the two L. plantarum strains have strong antioxidant potentials. The characteristics of the EPS produced make it a promising candidate for exploitation in the food industry.

Supplemental Information

Supplemental Information 1 TLC plate showing the monosaccharide composition of the EPS samples and Dry EPS produced on MRS-Sucrose modified media

TLC and dry EPS samples.

Click here for additional data file.

Data S1 Raw data

Click here for additional data file.

Additional Information and Declarations

Competing Interests

Author Contributions

DNA Deposition

Data Availability

The authors declare there are no competing interests.

Adekemi Titilayo Adesulu-Dahunsi, Kumaraswamy Jeyaram, Abiodun Isiaka Sanni and Kolawole Banwo conceived and designed the experiments, performed the experiments, analyzed the data, contributed reagents/materials/analysis tools, prepared figures and/or tables, authored or reviewed drafts of the paper, approved the final draft.

The following information was supplied regarding the deposition of DNA sequences:

L. plantarum YO175 and OF101 gene sequences are available at GenBank: numbers KU892395 and KU892393.

The following information was supplied regarding data availability:

The raw data are provided in the Supplemental Files.

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
