# Peer review of "Production of exopolysaccharide by strains of Lactobacillus plantarum YO175 and OF101 isolated from traditional fermented cereal beverage"

_PeerJ, doi:10.7717/peerj.5326_

## Round 0.1 · original submission · Major Revisions

Two reviewers have now commented on your paper. You will see that they are advising that you revise your manuscript. Please, give specific attention to the Statistical Analysis section.

·

Basic reporting

1. Lines 95-96, to the affirmation “Few works have been reported on EPS-producing LAB strains from cereals-based fermented food”, is there any reference?

2. The relevance of antioxidant activity of EPSs should be succinctly added to Introduction section, since this activity was tested in EPSs of present study. It can be mentioned that antioxidant activity of natural polysaccharides, including those of microbial EPSs, has gained great importance in recent decades since they are nontoxic antioxidants.

3. Figure 4, the contour plots which correspond to response surface three-dimensional plots present low resolution, which difficult the visualization. Please, improve their resolution.

Experimental design

1. The screening of optimal conditions for EPS production by isolated strains is not performed in food matrix, but in modified MRS broth. Thus, the optimal conditions and EPS yield found in MRS may be not reflected in food matrices. Therefore, the use of MRS broth in this study to evaluate the optimization conditions of EPS production by isolated LAB strains should be justified in Material and Methods section.

2. Line 119, the previous step performed to obtain the pellets mentioned in “The resulting pellets obtained were mixed with ultrapure water” should be described. Citations should not be used as a substitute for providing the details of a procedure.

3. Lines 133-134, please report your methods with sufficient detail so readers do not need to refer to other papers to understand how procedures were performed.

4. Lines 138-139, what column and run conditions of HPLC were employed? Were they the same used for molecular mass determination of EPSs? It need be specified.

5. Lines 139-140, please report your methods with sufficient detail so readers do not need to refer to other papers to understand how procedures were performed.

6. Lines 155-158. Describe, even if succinctly, the methods employed so readers do not need to refer to other papers to understand how procedures were performed.

Validity of the findings

1. Lines 145-150, the preliminary screening of cultivation condition and media composition for EPS production was performed to determine the significant factors for this production. However, what was the statistical analysis used to determine which factors are significant for EPS production? It was ANOVA? What significance level was considered for this statistical analysis? This information should be added in Statistical Analysis section.

2. Figure 6. It would be interesting to perform a statistical analysis comparing the antioxidant activities of EPSs studied with the control (ascorbic acid), well as to compare the antioxidant activities of EPSs between the different strains studied. The statistical analysis employed should be described in Statistical Analysis section. The statistical results should be added to graphs in Figure 6. The statistical analysis is an essential tool to determine if the differences found between the values are significant.

3. Table 2B, highly significant lack of fit (p-value = 0.0005) indicates that a new model is needed. The employed quadratic model for production of exopolysaccharide in Lactobacillus plantarum OF101 should not be complex enough to fit the data (important terms from the model such as interactions or quadratic terms should not have been included). From analysis of data, it seems that large residuals result from fitting the model should not has been the cause of lack of fit. I suggest evaluating other model that cans suitability to fit the data.

4. Figure 3, it is essential that be performed a statistical analysis with the data of Figure 3. It seems that there was no significant difference for EPS yield, for example, between the sucrose and lactose conditions, well as between the conditions of yeast extract and peptone, even so sucrose and yeast extract were reported by authors as optimal conditions for EPS yield when compared to others. The statistical analysis employed should be described in Statistical Analysis section. The statistical results should be added to graphs in Figure 3. The statistical analysis is an essential tool to determine if the differences found between the values are significant.

Additional comments

The manuscript titled, "Production of exopolysaccharide by strains of Lactobacillus plantarum YO175 and OF101 isolated from traditional fermented cereal beverage", proposed to optimize the production conditions of EPS by Lactobacillus plantarum strains isolated from ogi, a traditional fermented cereal beverage, followed by measurement of their antioxidant activities.

The paper was well organized, writter in good English and with a clear introduction. It was clearly stated how research fills an identified knowledge gap. The authors did a good job in results and discussion by clearly presenting results in tables and figures and comparing results with those described in literature. The weaknesses were the statistical analysis and the detailed poorly material and method section.

The reviewer recommends acceptance with major revisions.

1. Line 70, detail the GRAS term (Generally Recognized as Safe). The first mention in a text need be detailed.

2. Line 81, unclear sentence. Change “They form” by “It forms” if is referring to the ogi.

3. Line 86, typing error. Isolate the term Lactococcus of lactis term.

4. Lines 95-96, the statement is not very clear. I suggest to rewritter as follows: “Few works have been reported on EPS production ability by LAB strains isolated from cereals-based fermented food.”

5. Line 151 should be deleted. Lines 152-153 should be moved for the Statistical Analysis section.

6. Line 171, the authors report the figure S1 which is in Supplementary Material as Figure S2.

7. Line 176, the authors report the figure S2 which is in Supplementary Material as Figure S1.

8. In Discussion section, it is relevant compare the EPS yield found with data reported in literature. The results reported here were higher, lower or similar those reported in literature?

Reviewer 2 ·

Basic reporting

L71-75:One of the author's purpose is to evaluate the in vitro antioxidant activity of the EPS from L. plantarum strains. However, no background or rationale about the antioxidant activity of the EPS was done in the introduction section. I suggest the authors to describe here studies demonstrating antioxidant activity of EPS from lactobacillus.

L102: Change “characterize” to “evaluate”.

Experimental design

No comment.

Validity of the findings

L159: There are lacking information about the statistical analysis. Please describe all statistical tests performed in the manuscript.

L227. It is more interesting the authors show their results based on statistical tests instead of saying "the scavenging effects of the EPS increases as the concentration increased". Were the antioxidant capacity increases statistical significant? Please provide statistical sinificance information for all antioxidant analysis. Furthermore, I suggest include a correlation analysis in order to show the relationship between antioxidant activity and EPS concentration.

L282-285: The discussion on the antioxidant activity results should be improved... The authors can do it after making the statistical analysis as I have suggested above.

L291-292: Since no statistical analysis was done for the antioxidant analysis, the authors can´t state in their conclusion that the EPS produced by the two L. plantarum strains have significant antioxidant potentials.

Additional comments

The present study was developed with the purpose of improving the production of EPS from L. plantarum strains and to evaluate the antioxidant activity in vitro. Although the present work has a great scientific contribution, there are some points that need to be clarified in order to improve the manuscript.

---

## Round 0.2 · Minor Revisions

After careful consideration, we feel that the study has merit but does not fully meet PeerJ publication criteria as it currently stands. Therefore, we invite you to submit a revised version of the manuscript that addresses the points raised during the review process.

·

Basic reporting

No comment

Experimental design

No comment

Validity of the findings

1. Figure 3. The description of standard deviation among replicates not indicates if there is significant difference among the different treatments. The standard deviation measured the amount of variation or dispersion of set of data values within of a same treatment. The authors reported sucrose and yeast extract as optimal conditions for EPS yield when compared to others conditions. The question is: the differences in EPS yields observed for a same strain in different carbon sources (glucose, sucrose, lactose and galactose) are significant? The difference in EPS yields for a same carbon source (medium with glucose, for example) for different strains is significant? The same questions should be done for sucrose concentration, organic nitrogen, inorganic nitrogen, temperature, cultivation time, and pH. For the conjunct of data of EACH graph, I suggest to perform ANOVA two-way with significance level of 0.05. The statistics differences should be represented by symbols in each graph of Figure 3. In legend of figure should be briefly added information about the statistical analysis employed and description of symbols used.

2. Figure 6. The description of standard deviation among replicates not indicates if there is significant difference among the different treatments. The standard deviation measured the amount of variation or dispersion of set of data values within of a same treatment. For example, the authors reported that the reducing power of the ascorbic acid, EPS-YO175 and EPS-OF101 at 4 mg/mL concentration are (0.91˃0.41 ˃ 0.34). The question is: the reducing power of ascorbic acid, in this concentration, is significantly higher than of EPSs obtained from of different strains? In this concentration, the reducing power of EPS-YO175 is significantly higher than of EPS-OF101? Similar questions should be done for each concentration level of reductor agents. For the conjunct of data of EACH graph, I suggest to perform ANOVA two-way with significance level of 0.05. The statistics differences should be represented by symbols in each graph of Figure 6. In legend of figure should be briefly added information about the statistical analysis employed and description of symbols used.

Additional comments

1. In Supplementary Material, in legend "Dry EPS produced on MRS-Sucrose modified media", change the term Fig S2 by term Fig S1 in order to be in accordance with the description performed in manuscript.
2. In Supplementary Material, in legend "TLC plate showing the monosaccharide composition of the EPS samples", change the term Fig S1 by Fig S2 in order to be in accordance with the description performed in manuscript.
3. Lines 177-179, please, correct the legend referent to formula of the DPPH radical (%) scavenging activity: Ab = Absorbance of blank and Ac = Absorbance of control
4. Lines 336-337, your results should be discussed with those reported in literature.

Reviewer 2 ·

Basic reporting

No comment.

Experimental design

No comment.

Validity of the findings

No comment.

Additional comments

The authors have addressed all my comments in an adequate manner. I have no further comments.

---

## Round 0.3 · accepted · Accept

We are pleased to inform you that your manuscript has been judged scientifically suitable for publication and will be formally accepted for publication once it complies with all outstanding technical requirements.

#